# Motion-To-BMI: Using Motion Sensors to Predict the Body Mass Index of Smartphone Users

**DOI:** 10.3390/s20041134

**Published:** 2020-02-19

**Authors:** Yumin Yao, Ling Song, Jin Ye

**Affiliations:** 1School of Computer Science and Engineering, Central South of University, Changsha 410083, China; yaoyumin@csu.edu.cn; 2School of Information, Hunan Radio and Television University, Changsha 410004, China; 3School of Computer & Electronic Information, Guangxi University, Nanning 530004, China; jqian@gxu.edu.cn; 4Guangxi Key Laboratory of Multimedia Communications and Network Technology, Nanning 530004, China

**Keywords:** body mass index, prediction, motion sensors

## Abstract

Obesity has become a widespread health problem worldwide. The body mass index (BMI) is a simple and reliable index based on weight and height that is commonly used to identify and classify adults as underweight, normal, overweight (pre-obesity), or obese. In this paper, we propose a hybrid deep neural network for predicting the BMI of smartphone users, based only on the characteristics of body movement captured by the smartphone’s built-in motion sensors without any other sensitive data. The proposed deep learning model consists of four major modules: a transformation module for data preprocessing, a convolution module for extracting spatial features, a long short-term memory (LSTM) module for exploring temporal dependency, and a fully connected module for regression. We define motion entropy (MEn), which is a measure of the regularity and complexity of the motion sensor, and propose a novel MEn-based filtering strategy to select parts of sensor data that met certain thresholds for training the model. We evaluate this model using two public datasets in comparison with baseline conventional feature-based methods using leave-one-subject-out (LOSO) cross-validation. Experimental results show that the proposed model with the MEn-based filtering strategy outperforms the baseline approaches significantly. The results also show that jogging may be a more suitable activity of daily living (ADL) for BMI prediction than walking and walking upstairs. We believe that the conclusions of this study will help to develop a long-term remote health monitoring system.

## 1. Introduction

Obesity has become a widespread health problem worldwide, which may be related to an increased risk of diseases such as cardiovascular disease, diabetes, and stroke. The body mass index (BMI) is a simple and reliable index based on weight and height that is commonly used to identify and classify adults as underweight, normal, overweight (pre-obesity), or obese and was developed in the 19th Century by the Belgian statistician and anthropometrist Adolphe Quetelet. It is computed by its conventional definition that utilizes body height (cm) and mass (kg), $BMI=\frac{Mass}{Height^{2}}$. For adults, BMI falls into one of the following categories by the World Health Organization, presented in Table 1.

The conventional method to obtain BMI is usually to measure both body weight and height carefully; however, some patients who have a pre-obesity or obesity status may delay medical care because of concerns about disparagement by physicians and health care staff or a fear of being weighed [1]. Many users can use online applications on smart mobile terminals to obtain BMI. However, these applications require personal body data including height, weight, age, and sex, which are usually very sensitive topics. The measurement process requires the active participation of subjects to ensure the authenticity of the data, but improper usage of these personal body data easily leads to significant privacy and security threats.

Sensor-rich smartphones are popular worldwide. These built-in sensors, such as the camera, microphone, touch screen, and motion sensors, initially used for a phone’s enhancement, are now being used for a variety of sensing applications, and many aspects of human behavior and traits can be inferred, so some studies have begun to develop automatic measuring methods of BMI for long-term remote monitoring using smartphones on a large scale. Several existing studies have focused on learning based on human facial [2,3,4,5] and speech signals [6,7,8,9]. However, on the one hand, there are several potential risks to using these privacy-sensitive sensors such as cameras and microphones to collect users’ private data. The sensors mentioned above are also environmentally sensitive, so application scenarios are limited.

The human gait consists of the interaction between hundreds of muscles and joints in the body, and motion sensors can capture these and translate them into the characteristic patterns linked to their traits. In practice, it is more advantageous to obtain physical traits from motion sensors, due to the following reasons:(1)first and foremost, motion sensors are generally considered to be not highly privacy sensitive and more acceptable for users;(2)different from environmentally sensitive sensors, such as cameras and microphones, motion sensors always alleviate the limitations of the environment;(3)motion sensors are more popular than other sensors, not only in smartphones, but also in other smart devices, such as smart bracelets.

Motion sensor-based BMI prediction may be a great challenge. Firstly, the sensor data of the human gait have a comparatively low signal-to-noise ratio, i.e., sources that have BMI-relevant information affect signals less than sources that do not. Data from motion sensors are also multi-dimensional with a special temporal-spatial structure for conventional feature-based approaches [10,11]. Furthermore, the design of a specific feature extractor that transforms raw data into feature vectors relies on heuristic hand-crafted feature engineering and considerable domain expertise.

In the past decade, deep learning (DL) has achieved great success in many areas. A deep architecture with multiple layers is built up for automating feature design. Specifically, each layer in a deep architecture performs a nonlinear transformation of the outputs of the previous layer [12], so that DL can automatically make use of much more high-level and meaningful hidden features. The convolutional neural network (CNN) is a well-known DL model with the ability to learn complex, high-dimensional, nonlinear mappings from large collections of examples [13], and this makes them obvious candidates for image classification [14], speech recognition [15], and other recognition tasks related to time series [16,17,18]. Long short-term memory (LSTM) recurrent networks are also widely adopted for general-purpose sequence modeling, and they have proven stable and sturdy for long-range modeling dependencies in previous studies in such fields as audio analysis [19,20], video captioning [21,22], and sensor-based human activity recognition (HAR) [23]. Recently, the combination of CNNs and LSTM in a unified stack framework has already offered state-of-the-art results in speech recognition [24] and some motion sensor-based tasks [25].

The main contributions of the paper can be summarized as follows:(1)To the best of our knowledge, we are the first to design a hybrid deep neural network with a CNN-LSTM architecture to learn spatial features and temporal features from sensor data for identifying salient patterns related to the BMI.(2)We define motion entropy (MEn), which is a measure of the regularity and complexity of the motion sensor, and propose a novel MEn-based filtering strategy to select parts of these sub-sequences for training the prediction model.(3)We evaluate the hybrid deep neural network model with the MEn-based filtering strategy using two public datasets in comparison with baseline conventional feature-based approaches that have been applied to infer simple human traits from gaits [10,11]. Experimental results show that the proposed model significantly outperforms the baseline methods.(4)We also investigate which type of activities of daily living (ADLs) is more suitable for online BMI prediction.

The remainder of this paper is structured as follows: Section 2 reviews some related literature. We give a brief overview of the state of related works. In Section 3, we present the proposed hybrid deep neural network model and the corresponding MEn-based filtering strategy. Experimental results from the evaluation of two public datasets are presented in Section 4. Conclusions and a discussion are in Section 5.

## 2. Related Work

This section reviews closely related work in BMI prediction based on human facial and speech signals, inferring human traits from gaits and DL models based on CNNs, LSTM, and their combination.

### 2.1. BMI Prediction and Classification From Human Facial and Speech Signals

Recent studies have found that face images and speech signals have relations to BMI.

Wen et al. [2] developed a computational method to predict BMI from face images automatically. A promising result was obtained, which demonstrated the feasibility of developing a computational system for BMI prediction based on face images on a large scale. Kocabey et al. [3,4] used a state-of-the-art computer vision system to infer a person’s BMI from their social media profile photos. Polania et al. [5] employed a noisy binary search algorithm based on pairwise comparisons to exploit the ordinal relationship among BMI categories using facial images.

Lee et al. [6] suggested a novel method for BMI classification using speech signals and showed the possibility of predicting a normal status or an overweight status on the basis of voice and machine learning. Berkai et al. [7] reported the estimation of BMI status via speech signals using short-term cepstral speech feature extraction methods. A new method to predict BMI via speech signals (normal, obese, and overweight) using wavelet packet transform (WPT) and nonlinear entropy features was proposed by Berkai et al. [8]. Lim et al. [9] extracted energy and entropy features of speech, and both a k-nearest neighbour (kNN) and a probabilistic neural network (PNN) were used to measure the BMI of an individual.

### 2.2. Inferring Simple Human Traits from Gaits

Gait information captured by the smartphone’s built-in or wearable sensors can decode plenty of useful information about human traits. Weiss et al. [10] collected the accelerometer data of 70 participants for sex, height, and weight prediction tasks using conventional feature-based approaches. Sex prediction involved classifying a smartphone user as either male or female, while the height and weight prediction tasks included predicting the user’s height in inches and his/her weight in pounds. Their results indicated that, on average, WEKA’s instance-based (IB3) learner performed best and outperformed the straw man strategy of predicting height and weight. However, there was certainly room for future improvement. Riaz et al. [11] recorded accelerations and angular velocities of 26 subjects using wearable motion sensors attached at four locations (chest, lower back, right wrist, and left ankle) when performing standardized gait tasks. They then trained random forest classifiers in order to estimate soft biometrics (gender, age, and height).

### 2.3. CNNs, LSTM, and their Combination

CNNs and LSTM have shown promising results in some applications of multivariate time series, including, but not limited to, some works on motion sensors. CNNs have been applied to sensor-based human activity recognition (HAR) [16,17,18], which is the most relevant research topic of sensor-based gender recognition. LSTM has recently also shown promising results in a variety of sensor applications when there are sequential dependencies in time series. Through substantial scale experimentation in training [26], the procedures were analyzed for certain deep learning approaches for HAR, including deep LSTM networks. Zhao et al. [23] proposed a deep network architecture using residual bidirectional LSTM for human activity recognition (HAR), which showed improvements in both temporal (using bidirectional cells) and spatial (residual connections stacked) dimensions.

Recently, some effort has been made to combine aspects of CNN and LSTM architectures. Ordonez et al. [25] proposed a generic deep framework for HAR based on convolutional and LSTM recurrent units using multimodal wearable sensors. Their results showed that the framework could be applied to homogeneous sensor modalities, but multimodal sensors could also be fused to improve performance. Zhu et al. [27] presented a multimodal gesture recognition method based on 3D CNNs and LSTM networks, which firstly learned short-term features of gestures through a 3D convolutional neural network and then learned long-term spatiotemporal features by convolutional LSTM networks based on the extracted short-term spatiotemporal features.

### 2.4. To Quantify the “Complexity” of Physiological Signals

Quantifying the “complexity” of physiological signals in health has been the focus of considerable attention. Presently, there are three commonly used entropy-based algorithms for biological data: approximate entropy (ApEn) [28,29], sample entropy (SampEn) [30], and multiscale entropy (MSE) [31]. However, there is no simple entropy-based algorithm to tackle multiple types of motion sensor data at the same time.

## 3. Methodology

This section describes the proposed methodology to predict the BMI of smartphone users. We start with notations and definitions.

### 3.1. Notation and Definitions

#### 3.1.1. Sequence

A sequence *S* is an ordered list of multi-dimensional time series that are typically recorded in temporal order at fixed intervals. Given the *m*th subject, the sequence is $S_{m}$, and $T_{m}$ is the total number of intervals.
(1)$$S_{m}=\left\{d_{m}^{1},\ldots ,d_{m}^{i},\ldots ,d_{m}^{T_{m}}\right\}$$
$d_{m}^{i}$ denotes the *m*-th subject’s sensor recording (tri-axis accelerometer and tri-axis gyroscope) at the *i*-th sampling point and $i\in [1,T_{m}]$, as follows:(2)$$d_{m}^{i}=\left[\begin{array}{c} a_{m,x}^{i}\\ a_{m,y}^{i}\\ a_{m,z}^{i}\\ g_{m,x}^{i}\\ g_{m,y}^{i}\\ g_{m,z}^{i} \end{array}\right]$$

In this paper, the sequence $S_{m}$ will be segmented into a series of sub-sequences by a sliding windows strategy.

#### 3.1.2. Sub-Sequence

The de-facto standard workflow for processing sensor data in ubiquitous computing treats individual sub-sequences $x_{m}^{k}$ as statistically independent.

$x_{m}^{k}$, k∈[1,L], is the *k*th sub-sequence of the sequence $S_{m}$: (3)$$S_{m}=\left\{x_{m}^{1},\ldots ,x_{m}^{k},\ldots ,x_{m}^{L}\right\}$$

$L=\lfloor \frac{T_{m}-w}{\theta \;}\rfloor$, *w* is the length of each sub-sequence (in this paper, w=90, nearly 1 s), and θ is the step between the start intervals of two consecutive sub-sequences (θ=45, half of the window size). Concretely, $x_{m}^{k}$ has the sampling points from $d_{m}^{\left(k-1\right)\times \theta \;}$ to $d_{m}^{\left(k-1\right)\times \theta \;+w}$.
(4)$$x_{m}^{k}=\left\{d_{m}^{\left(k-1\right)\times \theta \;+1},\ldots ,d_{m}^{\left(k-1\right)\times \theta \;+w+1}\right\}$$

#### 3.1.3. Motion Entropy

For the sub-sequence $x_{m}^{k}$ with length *w*, MEn is an approximation of the negative mean natural logarithm of the conditional probability under some constraint of similarity tolerance.
(5)$$\mathrm{MEn}\left(x_{m}^{k},w,\xi ,r_{acc},r_{gyro}\right)=-\ln\left[\frac{B^{\xi +1}\left(r_{acc},r_{gyro}\right)}{B^{\xi }\left(r_{acc},r_{gyro}\right)}\right]$$

The parameters *w*, ξ, $r_{acc}$, and $r_{gyro}$ must be fixed for each calculation. *w* is the length of sub-sequence $x_{m}^{k}$; ξ is the length of the MEn template to be compared; and $r_{acc}$ and $r_{gyro}$ are the similarity tolerance of the accelerometer and the gyroscope for matching.

We proceed as follows:

$Acc^{i}$ and $Gyro^{i}$ are the vector magnitude of the tri-axis accelerometer and the gyroscope.
(6)$$Acc^{i}=\sqrt{{a_{x,t}^{i}}^{2}+{a_{y,t}^{i}}^{2}+{a_{z,t}^{i}}^{2}},Gyro^{i}=\sqrt{{g_{x,t}^{i}}^{2}+{g_{y,t}^{i}}^{2}+{g_{z,t}^{i}}^{2}}$$

$d_{m}^{i}$ denotes the *m*th subject’s sensor recording at the *i*th sampling point and $i\in [1,T_{m}]$.
(7)$$d_{m}^{i}=\left[\begin{array}{c} a_{m,x}^{i}\\ a_{m,y}^{i}\\ a_{m,z}^{i}\\ g_{m,x}^{i}\\ g_{m,y}^{i}\\ g_{m,z}^{i} \end{array}\right]\approx \left[\begin{array}{c} Acc_{m}^{i}\\ Gyro_{m}^{i} \end{array}\right]=n\left(i\right)$$

We convert the sub-sequence of raw sensor data into a vector amplitude sub-sequence. Given a sub-sequence $x_{m}^{k}$, which is the *k*th sub-sequence of the sequence $S_{m}$, *w* is the length of the sub-sequence, and θ is the step between the start intervals of two consecutive sub-sequences.
(8)$$x_{m}^{k}=\left\{d_{m}^{\left(k-1\right)\times \theta \;+1},\ldots ,d_{m}^{\left(k-1\right)\times \theta \;+w+1}\right\}$$
(9)$$\begin{array}{cc} x_{m}^{k} & \approx \left[\begin{array}{ccc} Acc_{m}^{\left(k-1\right)\times \theta \;+1} & \cdots & Acc_{m}^{\left(k-1\right)\times \theta \;+w+1}\\ Gyro_{m}^{\left(k-1\right)\times \theta \;+1} & \cdots & Gyro_{m}^{\left(k-1\right)\times \theta \;+w+1} \end{array}\right]\\ \; & =\left[\begin{array}{ccc} Acc_{m}^{\mu } & \cdots & Acc_{m}^{\mu +w}\\ Gyro_{m}^{\mu } & \cdots & Gyro_{m}^{\mu +w} \end{array}\right],\left(\mu =\left(k-1\right)\times \theta \;+1\right)\\ \; & =\{n(i):\mu \leq i\leq \mu +w\} \end{array}$$
{n(i):μ≤i≤μ+w} forms the w−ξ+1 matrices $N_{\xi }\left(j\right)$ for {j|μ≤j≤w−ξ+μ}, where $N_{\xi }\left(j\right)=\left\{n\left(j+k\right):0\leq k\leq \xi -1\right\}$ is the matrix of ξ data vectors from n(j) to n(j+ξ−1). The distance between two such matrices is defined as:(10)$$\begin{array}{cc} d[N_{\xi }\left(i\right),N_{\xi }\left(j\right)] & =\max\{|n(i+k)-n(j+k)|:0\leq k\leq \xi -1\}\\ \; & =\max\{|\left[\begin{array}{c} Acc_{m}^{i+k}\\ Gyro_{m}^{i+k} \end{array}\right]-\left[\begin{array}{c} Acc_{m}^{j+k}\\ Gyro_{m}^{j+k} \end{array}\right]|:0\leq k\leq \xi -1\}\\ \; & \approx \left\{\begin{array}{c} \max\{|Acc_{m}^{i+k}-Acc_{m}^{j+k}|:0\leq k\leq \xi -1\}\\ \max\{|Gyro_{m}^{i+k}-Gyro_{m}^{j+k}|:0\leq k\leq \xi -1\} \end{array}\right. \end{array}$$

Since the motion sensor matrix consists of two different types of sensor data, we study the maximum difference of their corresponding scalar components separately.

Let $B_{i}^{\xi }\left(r_{acc},r_{gyro}\right)$ be the number of matrices with the similarity tolerance $\max\{|Acc_{m}^{i+k}-Acc_{m}^{j+k}\left|\right\}\leq r_{acc}$ and $\max\{|Gyro_{m}^{i+k}-Gyro_{m}^{j+k}\left|\right\}\leq r_{gyro}$ of the accelerometer and the gyroscope.
(11)$$\begin{array}{cc} B_{i}^{\xi }\left(r_{acc},r_{gyro}\right) & =\frac{1}{(w-\xi )\times 2}\mathrm{num}\{\max\{|Acc_{m}^{i+k}-Acc_{m}^{j+k}\left|\right\}\leq r_{acc}\;\\ \; & +\max\{|Gyro_{m}^{i+k}-Gyro_{m}^{j+k}\left|\right\}\leq r_{gyro}\},\\ \; & i=1,2,\cdots w-\xi +1,i\neq j \end{array}$$

We then average all $B_{i}^{\xi }\left(r_{acc},r_{gyro}\right)$ values of the parameter *i*, which is the probability of the self-similarity of the vector magnitude series with the number ξ.
(12)$$B^{\xi }\left(r_{acc},r_{gyro}\right)=\frac{1}{w-\xi +1}\sum_{i=1}^{w-\xi +1}B_{i}^{\xi }\left(r_{acc},r_{gyro}\right)$$

$B^{\xi +1}\left(r_{acc},r_{gyro}\right)$ can obtain the similarly, which is the probability of the self-similarity of the vector magnitude series with the number ξ+1.

In practice, Equation (Equation 5) can be abbreviated as:(13)$$\mathrm{MEn}\left(x_{m}^{k},\psi \times \bar{std}\right)=-\ln\left[\frac{B^{\xi +1}\left(\psi \right)}{B^{\xi }\left(\psi \right)}\right]$$
where $\bar{std}$ stands for the mean of the vector magnitude series’ standard deviation, which should be taken over a related large dataset. Generally, we take the value of tolerance parameter ψ=0.01,0.05,0.1.

#### 3.1.4. MEn-Based Filtering

We propose a MEn-Based filtering algorithm to select parts of these sub-sequences adjusted by threshold parameter θ and tolerance parameter ψ, for training the prediction model. The setting of these two parameters (Algorithm 1) will be discussed in Section 4.5.2.
**Algorithm 1:** MEn-based filtering algorithm.
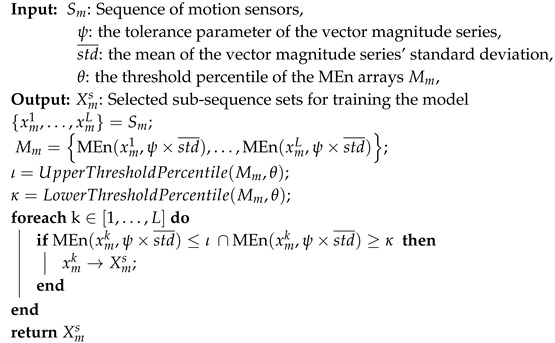


### 3.2. The Proposed Hybrid Deep Neural Network Model

In this section, we introduce the proposed hybrid deep neural network for training the BMI prediction model shown in Figure 1. The key components include four major modules: a transformation module, a convolution module, an LSTM module, and a fully connected module.

First, the sequence of the motion sensors was divided into many sub-sequences by sliding windows, and every sub-sequence was labeled with a BMI value. Parts of the sub-sequences were then selected by an MEn-based filtering strategy. Finally, the selected sub-sequences were used as the input of the DL network for training a prediction model.

#### 3.2.1. The Transformation Module

The recordings of the motion sensor were a batch of multi-dimensional (a 6D, 3-axis accelerometer and a 3-axis gyroscope) sequences, as shown in Section 3.1.1, which may have different properties and correlations with respect to one another. The proposed pipeline was a popular way of using DL models for multi-dimensional time series prediction purposes. This method consisted of modifying the de-facto standard processing workflow in accordance with how it has been widely adopted by the community of ubiquitous computing. The main difference is that we propose a novel MEn-based filtering strategy to select sub-sequences that satisfy certain thresholds.

This module applied four steps to transform raw recordings into suitable data representations for further processing. In what follows, we discuss each of them them.

Step 1: Resample and interpolate

Since different sensors may have different sampling rates, to guarantee that the data of an accelerometer and a gyroscope can be processed simultaneously, the sampling rates of the two sensors must be unique. Hence, we linearly resampled and interpolated all sensors’ recordings to the sampling rate, which was the starting point for this module.

Step 2: Window Sliding 

We used a fixed-size sliding window strategy to split the data, which was adapted to segment the sequence of the time series into a collection of sub-sequences, as shown in Section 3.1.2.

Step 3: Normalization 

The data in a given window were normalized to the range 0...1by the Z-score method. By normalization, the data from different datasets were standardized without changing their distribution.

Step 4: Data Filtering 

As mentioned above, in Section 4.5.2, we defined motion entropy (MEn), which was a measure of the regularity and complexity of the motion sensor, and proposed an MEn-based filtering strategy to select parts of these sub-sequences for training the model. By experiments, in Section 3.1.4, we showed that only the data of motion sensors that were consistent with the balance of the signal’s regularity and complexity were likely to obtain more physiological characteristics, including obesity.

#### 3.2.2. The Convolution Module

In the convolution module, we applied an AlexNet-like [14] CNN structure on each branch independently. The transformation module generated multivariate time series from motion sensors, and 2-dimensional convolution and max pooling operations were shown to be desirable for capturing local dependency between different channels, as well as local dependency between time steps [32].

The 2-dimensional convolution layer *l* operation was performed by calculating a feature map $c_{i,j}^{l,M}$:(14)$$c_{i,j}^{l,m}=\varphi \left(\sum_{m^{\prime }=1}^{M^{\prime }}\sum_{x=1}^{X}\sum_{y=1}^{Y}w_{x,y,m^{\prime }}^{l-1,k}z_{i+x-1,j+y-1}^{l-1,M^{\prime }}+b^{l-1,m}\right)$$
where *X* and *Y* are the size of the 2D convolution kernel running over space and time, respectively, $M^{\prime }$ is the number of feature maps in the convolutional layer (l−1), $w^{l-1,m^{\prime }}\in R^{X}\times Y\times M^{\prime }$ is a local filter weight tensor, and $b^{l-1,k}\in R$ is a bias.

The max pooling operation produces:(15)$$c_{i,j}^{l,k}=\max_{\left(i-1\right)S+1\leq i^{\prime }\leq iS,\left(i-1\right)T+1\leq i^{\prime }\leq iT}c_{i^{\prime },j^{\prime }}^{l-1,k}$$
where *S* and *T* are the size of the pooling kernel running over space and time, respectively.

We designed three 2D convolutional layers and used a series of rectangle convolution kernels, which are different from square convolution kernels usually used in image recognition [14,33].

The number of feature maps and the size of convolution kernel could be adjusted, depending on the datasets. In our case, the settings mentioned above worked well, so we used them here. For each convolutional layer, a rectifying linear unit (ReLU) was added as the activation function with batch normalization (momentum = 0.99, epsilon = 0.001) and was often followed by a max-pooling layer to increase the robustness of the extracted features [34].

The output of the 2D convolution and max pooling layer was a 3D tensor of shape (space, time, and channel). The space and time dimensions tended to shrink as one went deeper into the network. The number of channels was controlled by the argument passed to the 2D convolution layers. The current output was a 3D tensor, whereas the input of the following LSTM module was a 1D vector, so the 3D outputs of each branch had to be flattened to 1D and merged.

#### 3.2.3. The LSTM Module

As part of the hybrid network, the convolution module was designed to extract spatial features, and the LSTM module was expected to learn long-term dependencies.

The major innovation of LSTM was its memory cell, which could be stored in, written to, or read from neurons in the network and acted much like a computer’s memory. The updating of an LSTM unit at every time step *t* is described as follows, from Equations (Equation 16)–(Equation 19).

At time t, whether data are written to the LSTM cell depends on input gate $i_{t}$; whether data are erased from the LSTM cell depends on input gate $f_{t}$; and how much of an LSTM cell is revealed depends on input gate $o_{t}$. These gates protect and control the LSTM cell state $C_{t}$. σ is the component-wise logistic sigmoid function; $b_{x}$ are the bias vectors; and $W_{x}$ represents the weight matrices. $x_{t}$ is the input of the layer; $h_{t}$ is the layer output; and $C_{t}$ is the cell output state.
(16)$$i_{t}=\sigma \left(W_{i}\cdot \left[h_{t-1},x_{t}\right]+b_{i}\right)$$
(17)$$f_{t}=\sigma \left(W_{f}\cdot \left[h_{t-1},x_{t}\right]+b_{f}\right)$$
(18)$$o_{t}=\sigma \left(W_{o}\left[h_{t-1},x_{t}\right]+b_{o}\right)$$
(19)$$C_{t}=f_{t}*C_{t-1}+i_{t}*{\tilde{C}}_{t}$$
(20)$${\tilde{C}}_{t}=\tanh\left(W_{c}\cdot \left[h_{t-1},x_{t}\right]+b_{c}\right)$$

Multiple LSTMs were designed to be stacked and temporally concatenated to form more complex structures.

#### 3.2.4. The Fully Connected Module

We passed the output of the LSTM module to the fully connected module. As shown in Figure 1, this layer was the same as a standard multilayer perceptron neural network that mapped the latent features into the output classes we wanted to discriminate. With the fully connected layers, we combined these features together to create a regression model. The activation function of the output layer was a linear function because we wanted to obtain a continuous BMI value. Finally, we compiled the network with the mean squared error (MSE) loss function, which is a widely used loss function for regression problems.

## 4. Experiments

### 4.1. Dataset Description

In the experiments, we considered two public datasets, MobiAct [35] and Motion-Sense [36]. We proposed a deep learning model with other traditional feature-based approaches to check them.

In the following, we provide an introduction of MobiAct and Motion-Sense. Unlike other public datasets that require the smartphone to be rigidly placed on the human body and with a specific orientation, the datasets we selected were from devices that were freely located in pants pockets in a random orientation. These data corresponded to daily life, which also distinguished them from other data that were acquired from one or more wearable sensors banded on certain sensitive parts of the human body.

#### 4.1.1. MobiAct

The MobiAct dataset consisted of recordings from 67 subjects. The average age of subjects was 25.19, the average height 175.75 cm, and the average weight 76.80 kg. Motion sensor data were recorded by a Samsung Galaxy S3 with the LSM330DLC inertial module.

#### 4.1.2. Motion-Sense

The Motion-Sense dataset consisted of recordings from 24 subjects (14 men and 10 women) with a balance of gender. The subjects’ age spanned between 18 and 46, the height from 161 to 190 cm, and the weight from 48 to 102 kg. For each participant, an iPhone 6s was located in the pants pocket freely and in a random orientation. The motion sensors were logged by SensingKit, which included accelerometer, gravity sensor, gyroscope, and attitude sensor data.

#### 4.1.3. BMI values and the Nutritional Status of the Two Datasets

The BMI values of the two benchmark datasets are shown in Figure 2. According to the conversion method of Table 1, the nutritional status of the subject associated with BMI is shown in Figure 3. We found that the normal weight and pre-obesity population accounted for a large proportion, which was mainly related to the data samples collected from the university campus, and the average age of volunteers was less than age 30.

### 4.2. Comparison with Existing Methods

We compared our proposed hybrid deep neural network model with some state-of-the-art feature-based methods using WEKA [37], which included a k-nearest neighbor algorithm (kNN) that is called an instance-based learner (IBk) in WEKA [10], a support vector machine (SVM), and a C4.5 decision tree that is called J48 in WEKA [10]. Among them, the data contained in one example duration were converted into a single example, described by 43 features [10], which were variations of six essential features, including the average acceleration value, the standard deviation, the average absolute difference, the average resultant acceleration, the time between peaks, and the binned distribution.

IBk uses a distance measure to locate k “close” instances in the training data for each test instance and uses those selected instances to make a prediction.

SVM with radial the basis function (RBF) kernel was used as the classifier.

J48 is an algorithm used to generate a decision tree that can be used for prediction, and for this reason, J48 (C4.5) is often referred to as a statistical classifier.

### 4.3. Leave-One-Subject-Out Cross-Validation

Suppose a dataset with *N* subjects. For each experiment, we used N−1 subjects’ sensor data for training and the remaining subject’s sensor data for testing. At first, in LOSO, the subjects ${\left\{S_{n}\right\}}_{n=1}^{N}$ were partitioned into *N* groups. The samples were then partitioned by the groups into *N* sub-samples ${\left\{D_{n}\right\}}_{n=1}^{N}$ of the *N* sub-samples. A single sub-sample $D_{test}$ was retained for testing the model, and the remaining N−1 sub-samples $D_{train}$ were used as the training data. Finally, the CV process was then repeated *N* times, with each of the *N* sub-samples used exactly once as the validation data.

### 4.4. Evaluation Measures

To have a fair comparison, we used two kinds of regression measures, namely, mean absolute error (MAE) and root mean squared error (RMSE), to evaluate the performance of different methods on the test data in the experiments.

(1)Mean absolute error (MAE):The MAE is the absolute value of the difference between the predictions and the targets (L1 norm). It is a linear score, which means that all the individual differences are weighted equally in the average.
(21)$$MAE=\frac{1}{N}\sum_{i}^{N}\left|y_{i}-{\hat{y}}_{i}\right|$$(2)Root mean squared error (RMSE):The RMSE is a quadratic scoring rule that also measures the average magnitude of the error (L2 norm). It’s the square root of the average of the squared differences between the prediction and the actual observation. The RMSE is more sensitive to outliers, and MAE is more stable.
(22)$$RMSE=\sqrt{\frac{1}{n}\sum_{i=1}^{n}{\left(y_{i}-{\hat{y}}_{i}\right)}^{2}}$$

### 4.5. Results

In this section, we evaluate the performance of the proposed prediction model on two datasets and compare it with three other feature-based methods. Additionally, we verify the effectiveness of the MEn-based filtering strategy, and we discuss what types of ADLs might be suitable for the prediction of BMI.

#### 4.5.1. Experiments without Data Filtering

To demonstrate the performance of our proposed hybrid deep neural network (CNN-LSTM), we used LOSO cross-validation on MobiAct and Motion-Sense.

Firstly, to give a reliable performance comparison, we trained the model for a relative optimal network structure and hyper-parameters. Since CNN-LSTM is deep in both the spatial domain and the time domain, there were different structures that may have an impact on the performance of the model. For example, considering only the number of network layers, the number of convolutional layers may affect the ability of the model to learn the spatial structures. The number of recursive hidden layers may affect the intensity of the model’s learning of temporal relationships. The number of fully connected hidden layers may change the learning feature transformation. The optimization of hyper-parameters also improved the overall performance and generalization capacity of the model. Tuning hyper-parameters for the DL model, however, was a time consuming and challenging task because there were numerous parameters to be configured. In this work, we used the grid search method to determine the optimal hyper-parameters of the proposed model. Six common hyper-parameters, namely the optimizer, the learning rate, the number of epochs, the batch size, the dropout rate, and the regularizer, were optimized. To implement a grid search more efficiently, the search spaces of hyper-parameters were manually selected in the initial stage after experiments on the datasets. The optimized parameters, their search spaces, and their determined optimal values are presented in Table 2.

Secondly, the results presented in Table 3 and Table 4 show that our proposed hybrid deep neural network (CNN-LSTM) with an optimal structure led to significant performance improvements compared with conventional feature-based models in all cases.

Finally, both Table 3 and Table 4 show that jogging was the most suitable activity for BMI prediction, based on a comparison of four ADLs. In practice, an acceptable BMI prediction value could also be obtained while walking and walking upstairs.

#### 4.5.2. Experiments with Data Filtering

To verify the effectiveness of our proposed MEn-based filtering strategy, we performed comparisons among three different data filtering settings by threshold parameter θ and tolerance parameter ψ, as mentioned in Section 3.1.4. The experimental results showed that the entropy filtering strategy improved the performance of the CNN-LSTM model by about 10%, if the parameters were set properly.

Table 5 shows that the MAEs of the BMI were 2.461 ± 1.000 in the jogging state in the MobiAct dataset and 3.137 ± 1.300 in the jogging state in the Motion-Sense dataset. This indicated that the error of our proposed model was low, considering the wide range of BMI from 17 to 35, visually shown in Figure 2.

As shown in Figure 3, both datasets were quite unbalanced, as subjects in the underweight and obesity categories are scarcity. Table 6 shows the total number of sub-sequences (samples) in the jogging state selected by data filtering. Taking Motion-Sense as an example, the normal weight category with the threshold parameter θ = 0.1 contained the highest number of sub-sequences (samples), which was about 57.6% of the whole set. The pre-obesity category was the second largest, which contained 431 sub-sequences (samples). These two categories possessed about 92.5% of the whole set.

The MAE was defined as the average of the absolute errors between the predicted BMIs and the ground-truth BMIs. Figure 4 shows the MAEs of our proposed CNN-LSTM model in most BMI categories: underweight, normal weight, pre-obesity, and obesity, with threshold parameter θ = 0.1 and tolerance parameter ψ = 0.05 in the jogging state. As a result, we obtained a higher accuracy, 94.8% ± 1.5%, in predicting BMI according to the recordings of motion sensors in the normal and pre-obesity BMI categories. The lower prediction accuracies in the underweight and obesity categories were probably because of the limited sub-sequences in both training and testing, as shown in Table 6.

## 5. Conclusions and Discussion

In this paper, we proposed a hybrid deep neural network with a novel MEn-based filtering strategy for predicting the BMI of smartphone users using built-in motion sensors only. We also have shown that the accuracy of our proposed prediction model is the highest in a jogging state, and an acceptable BMI prediction value can also be obtained while walking and walking upstairs, by extensive experiments on two public datasets. We believe that the conclusions of this study will help to develop a long-term remote BMI monitoring system. The so-called long-term remote monitoring refers to the collection of a few motion sensor data every day for many years to trace a person’s physical conditions, which is different from the short-term real-time recognition of human activity.

Despite the progress made in this work, sensor-based BMI predictions remain challenging. The performance of deep learning models still heavily depends on labeled samples. Obtaining enough activity labels is expensive and time consuming. Therefore, in order to predict the BMI value corresponding to sensor data without activity labels, unsupervised HAR is urgent. We will try to use transfer learning in subsequent research to perform data annotation by leveraging labeled data from other auxiliary domains.

## Figures and Tables

**Figure 1 sensors-20-01134-f001:**
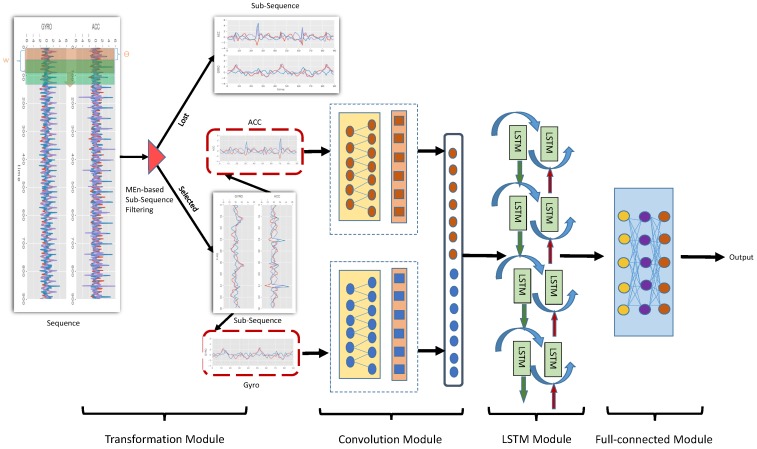
Overall architecture of the proposed hybrid deep neural network.

**Figure 2 sensors-20-01134-f002:**
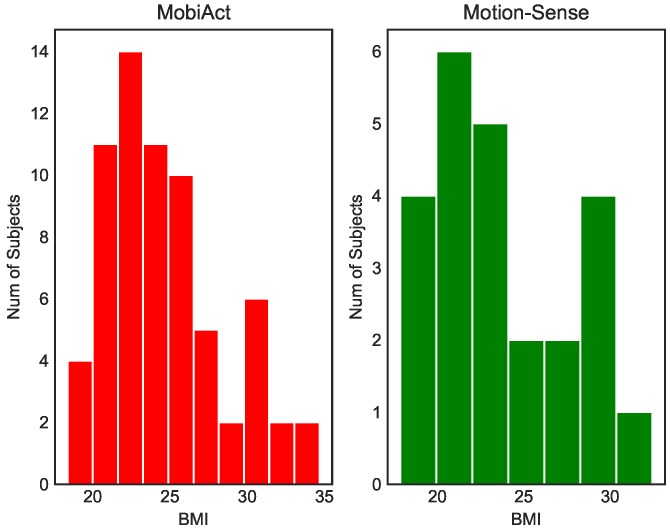
The BMI of the two datasets.

**Figure 3 sensors-20-01134-f003:**
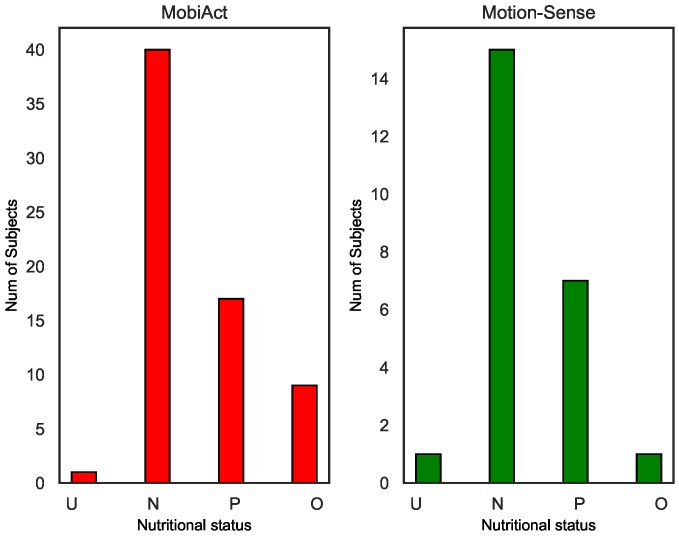
The nutritional status related to BMI of the two datasets. U: underweight; N: normal weight; P: pre-obesity; O: obesity.

**Figure 4 sensors-20-01134-f004:**
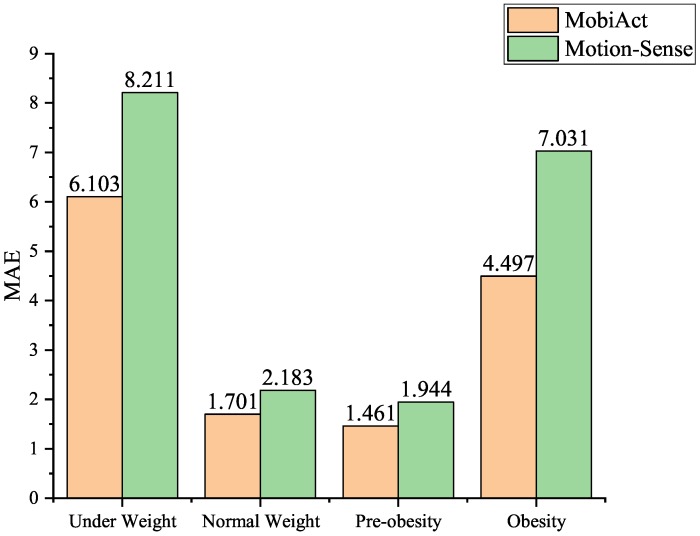
The MAEs in different BMI categories with optimal parameters.

**Table 1 sensors-20-01134-t001:** Nutritional status.

BMI	Nutritional Status
18.5	Underweight
18.5–24.9	Normal weight
25.0–29.9	Pre-obesity
30.0–34.9	Obesity Class I
35.0–39.9	Obesity Class II
40	Obesity Class III

**Table 2 sensors-20-01134-t002:** Hyper-parameters of the CNN-LSTM.

Parameter	Search Space	Optimal Values
Optimizer	(Rmsprop, Adam, Sgd)	Adam
Learning Rate	(0.0001, 0.001, 0.01)	0.001
Epochs	(50, 100, 200, 300)	200
Batch Size	(10, 20, 30, 40)	20
Dropout Rate	(0, 0.2, 0.5)	0.5
Regularizer	(L1, L2)	L2

**Table 3 sensors-20-01134-t003:** The experimental results in MobiAct without data filtering.

Dataset	ADL	Model	MAE	MSRE
MobiAct	Jogging	CNN-LSTM	1.836	2.547
		IBk	2.973	4.192
		SVM	2.497	7.156
		J48	2.314	5.272
	Walking	CNN-LSTM	2.190	3.804
		IBk	2.810	4.962
		SVM	3.293	6.910
		J48	3.455	5.040
	Walking upstairs	CNN-LSTM	1.919	2.978
		IBk	2.039	3.746
		SVM	3.926	5.804
		J48	2.421	6.002
	Walking downstairs	CNN-LSTM	3.203	4.771
		IBk	4.902	6.296
		SVM	5.614	7.472
		J48	5.403	6.921

**Table 4 sensors-20-01134-t004:** The experimental results on Motion-Sense without data filtering.

Dataset	ADL	Methods	MAE	MSRE
Motion-Sense	Jogging	CNN-LSTM	2.319	3.817
		IBk	3.213	4.791
		SVM	3.907	7.143
		J48	5.078	9.037
	Walking	CNN-LSTM	2.976	4.301
		IBk	2.401	4.097
		SVM	4.723	7.072
		J48	6.002	9.426
	Walking upstairs	CNN-LSTM	2.607	3.572
		IBk	2.319	3.491
		SVM	3.193	4.147
		J48	5.733	6.710
	Walking downstairs	CNN-LSTM	4.109	6.018
		IBk	5.312	8.830
		SVM	5.901	8.012
		J48	7.517	9.521

**Table 5 sensors-20-01134-t005:** The experimental results in the jogging state with data filtering.

Dataset	Threshold Percentile	Measures	Tolerance Parameter		
			ψ = 0.01	ψ = 0.05	ψ = 0.1
MobiAct	θ = 0	MAE	1.836		
		MSRE	2.547		
	θ = 0.05	MAE	2.239	1.930	2.861
		MSRE	3.418	2.806	3.726
	θ = 0.1	MAE	1.793	1.623	2.156
		MSRE	2.541	2.410	2.962
	θ = 0.25	MAE	3.352	2.304	3.449
		MSRE	4.917	3.705	5.024
Motion-Sense	θ = 0	MAE	2.319		
		MSRE	3.817		
	θ = 0.05	MAE	2.613	2.107	3.445
		MSRE	2.541	2.410	2.962
	θ = 0.1	MAE	2.268	2.005	3.574
		MSRE	3.773	3.049	4.218
	θ = 0.25	MAE	4.530	3.631	5.108
		MSRE	5.311	4.503	5.919

**Table 6 sensors-20-01134-t006:** Total number of sub-sequences (samples) in the jogging state selected by data filtering.

Categories	MobiAct				Motion-Sense			
	θ = 0	θ = 0.05	θ = 0.1	θ = 0.25	θ = 0	θ = 0.05	θ = 0.1	θ = 0.25
Underweight	43	40	34	22	67	58	51	35
Normal weight	1376	1194	1063	691	904	824	712	467
Pre-obesity	895	812	728	467	561	497	431	298
Obesity	603	538	474	310	58	47	40	26

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
