# Peer review of "Motion-To-BMI: Using Motion Sensors to Predict the Body Mass Index of Smartphone Users"

_sensors, 2020, doi:10.3390/s20041134_

Round 1
Reviewer 1 Report
The examples of BMI prediction demonstrate that previous attempts have an unacceptably large mean absolute error of between 3 (normal, underweight) and 10 (obese) BMI units. The conclusions from previous literature do not support BMI prediction from facial images to be of any clinical or diagnostic use. The idea of using motion sensors to estimate BMI and specifically obesity is questionable given the overwhelming evidence that those who are overweight may do as much or more exercise than those of normal BMI. Figures 6 and 7 ought to provide the regression plots, regression coefficients and SEEs to establish the accuracy of the prediction. The conclusions contain no estimates of prediction error. The Tables indicate that the MAE is in the order of 2.5 to 7.5 BMI units and that the R2 values are mostly less than 10% leaving almost 90% of the factors affecting prediction unexplained. To conclude that a jogging assessment via smartphone is the best predictor of BMI leaves so many fundamental factors unchallenged that it is highly questionable as to whether this approach is worthwhile as anything more than a simple motion sensor with unacceptably large prediction errors for physical traits.Author Response
Please see the attachment.

Reviewer 2 Report
This paper presents a novel deep-learning based BMI prediction through motion sensors. The problem is interesting and useful in life. But it might be better to see how the system will work on the dataset without activity labels and whether the prediction accuracy is good enough for medical purposes.
Strength
- Novel problem and convincing motivation
- Extensive and solid evaluation and comparison.
Weakness
- Not sure whether the system works for unlabeled data.
- Not sure whether the prediction is accurate enough for medical purposes.
Comments
- The paper concludes that jogging is the most suitable activity for BMI prediction. It might be true from the dataset with labels. But how can the system tell the type of activity for unlabeled data, e.g., new data from a user? No discussion about how to predict users' activity in the paper. If the users are not jogging, do you still run the prediction?
- How accurate the BMI predictions are? Are they accurate enough? Although in Fig. 6 and Fig. 7 the authors compare the prediction value vs. true values, it not clear how many samples are correctly classified according to nutritional status labels in Table 1. Giving the prediction accuracy will be helpful to evaluate whether the system will be useful in a practical scenario.
- Fig 6 and 7 are not self-contained and descriptions are also lacked. It is not clear what data are plotted in these two figures. Is it jogging data from both datasets under the highlighted parameters in Table 5? Please add legends/captions or descriptions to clarify the dataset and parameters used in these two figures. In addition, for Fig. 5, exchange 'u' and 'o' labels in the right part to be consistent with the left part.
- What do you mean by 'sensor data are complexity enough' throughout the paper? Please be specific.
- Some typos:
- Line 35: 'to using' -> 'to use'
- Line 233: 'a AlexNet' -> 'an AlexNet', 'independent' -> 'independently'
Reviewer 3 Report
This manuscript presents a deep learning-based approach that aims to automatically evaluate the Body Mass Index of a subject, from the processing of the motion data generated by sensors (accelerometer and gyroscope) onboard a smartphone carried in a random position inside pockets.
The title of this work suggests a quite innovative and interesting topic, however some concerns arise when reading this manuscript:
1) motivation of the study seems weak, as there are many smart scale models, connected to the internet, that are easy to use and more precise than the proposed system;they may be used to get the BMI automatically on someone's smartphone, through an app, after having set own body height, age and sex. So, statements in lines 26-28 should be strongly revised;
2) the link between the application of DL to motion sensor data and the BMI is not explained. What is the correlation between them?
3) as BMI doesn't change on a short time, it is not necessary to monitor it continuously. Continuous acquisition of motion sensor data is indeed necessary in human activity recognition;
4) lines 230-231: as the method proposed is heuristic, authors should give more details about the experiments performed and how the thresholds have been determined to better set up the algorithms;
5) Figures 2 and 3: these graphs are almost useless as the labels are unreadable and the quantity on the horizontal axis is missing;
6) in Figure 4, using half values on the vertical scale in the right graph makes no sense. Additionally, use the same axis limits as in the left graph and measurement unit on the horizontal axis is missing in both the graphs;
7) in Figures 6 and 7 please specify what dataset has been used;
8) line 388: please avoid using the term "diagnosis" as it would require clinical validation, while, in your case, you are presenting results on a population that is not statistically significative nor accounts for different health statuses. The datasets are quite unbalanced as the number of subject with exceeding BMI values is very short;
9) when expressing a measurement, always put a space between the value and the measurement unit. Pay attention to the correct definition of the BMI and related measurement unit (height must be given in cm and not in m).
It is mandatory that authors perform an accurate proofreading of their manuscript, as there are many typos, grammar errors, and also sentences that are totally not clear to understand. Many of these issues are highlighted in the attached PDF file.

Round 2
Reviewer 3 Report
Authors have addressed the comments raised after the first review round in a quite acceptable way. Still, the motivations of the study are not totally convincing, also because in the end it emerges how recognizing BMI from sensor data is not such an efficient approach and further investigations are needed.
Anyway, the manuscript has been improved overall, with respect to the initial submission.
It is still requested that authors perform an accurate proofreading of their manuscript, as there are many typos, grammar errors, and also sentences that are difficult to understand.
Author Response
Please see the attachment

This manuscript is a resubmission of an earlier submission. The following is a list of the peer review reports and author responses from that submission.